# Prediction of improvement after extended thymectomy in non-thymomatous myasthenia gravis patients

**Mitsuteru Yoshida**[1]*, **Kazuya Kondo**[1], **Naoko Matsui**[2], **Yuishinn Izumi**[2], **Yoshimi Bando**[3], **Michihiro Yokoishi**[4], **Kouichirou Kajiura**[1], **Akira Tangoku**[1]

1 Department of Thoracic, Endocrine Surgery and Oncology, Institute of Health, Bioscience, Graduate School, University of Tokushima, Tokushima, Japan, 2 Department of Neurology, Institute of Health Bioscience, Graduate School, University of Tokushima, Tokushima, Japan, 3 Tokushima University Hospital Division of Pathology, Tokushima, Japan, 4 Tokushima University Hospital Division of Radiology, Tokushima, Japan

* mitsuteru@tokushima-u.ac.jp

## Abstract

### Background

It is popularly believed that myasthenia gravis (MG) patients show acetylcholine receptor antibody (AChRAb) production associated with the thymus (germinal centers, approximately 80%). It has been suggested that thymectomy can remove the area of autoantibody production. This study aimed to determine whether the solid volume of the thymus calculated using three-dimensional (3D) imaging could be used to predict the efficacy of thymectomy. Additionally, the study assessed the relationships of the solid volume with germinal centers, change in the serum AChRAb level, postoperative MG improvement, and prednisolone (PSL) dose reduction extent.

### Methods

This retrospective study included 12 consecutive non-thymomatous MG patients (9 female and 3 male patients), who underwent extended thymectomy at our institution over the last 10 years. The mean patient age was 43.3 ± 14.2 years (range, 12–59 years). The study assessed the number of germinal centers per unit area, change in the serum AChRAb level, postoperative MG improvement, PSL dose reduction extent, and solid volume of the thymus.

### Results

The number of germinal centers per unit area was significantly correlated with the solid volume of the thymus. The PSL dose reduction extent tended to be correlated with the solid volume.

### Conclusions

Our findings suggest that the solid volume of the thymus can possibly predict steroid dose reduction. Additionally, the solid volume of the thymus in 3D images is the most important indicator for predicting the efficacy of extended thymectomy.

**Data Availability Statement:** All relevant data are within the manuscript.

**Funding:** The authors received no specific funding for this work.

**Competing interests:** The authors have declared that no competing interests exist.

## Introduction

In 1939, Blalock reported that a 21-year-old woman with myasthenia gravis (MG) experienced symptomatic improvement after resection of a cystic thymic tumor [1]. The efficacy of thymectomy for improving outcomes in patients with non-thymomatous MG is still being studied, although it has been used in clinical practice for over 60 years [2–5]. Carolina Barnett et al. used novel statistical techniques to assess observational data and reported that thymectomy is associated with a high probability of achieving remission or a minimal-manifestation status and achieving freedom from prednisone use when compared to the findings in controls [5]. Recently, the MGTX Study Group found that thymectomy improved clinical outcomes over a 3-year period in patients with non-thymomatous MG [5, 6]. Further prospective studies are needed to assess the benefits of thymectomy, and if its benefits are confirmed, cost-effectiveness studies will be needed to help further understand its role in the management of MG patients [5]. It is believed that the germinal centers of the thymus are responsible for the selection of B cells capable of producing acetylcholine receptor antibodies (AChRAbs) and that selected B cells act as a source of AChRAbs in peripheral regions. Therefore, it has been suggested that thymectomy can remove the area responsible for AChRAb production. There is a high probability that MG patients have AChRAbs from the thymus, including germinal centers (about 80%) [7, 8] and thymomas (20–25%) [9, 10]. Almost all germinal centers of the thymus are considered to exist in the solid area (thymic tissue) on computed tomography (CT). Therefore, we focused on the solid area as the location of AChRAb production.

The present study aimed to determine whether the solid volume of the thymus, calculated using three-dimensional (3D) imaging, could be used to predict the efficacy of thymectomy. Additionally, the study assessed the relationships of the solid volume with germinal centers, change in the serum AChRAb level, postoperative MG improvement, and prednisolone (PSL) dose reduction extent.

## Methods

The study design was approved by the appropriate ethics review board of Tokushima University Hospital (approval number: 3383–1). This study design is a retrospective review of patient medical reviews. The specific source of the medical records is the electronic medical records of Tokushima University Hospital. We could contact some patients; hence, written informed consent could be obtained from these participants. On the other hand, for the patients which we could not contact, we published the information disclosure documents on the home page of Tokushima University Hospital. The IRB specifically waived the requirement for informed consent for patients who could not be contacted. We determined the abovementioned procedures about informed consent according to the Japanese governmental guidelines and have received approval from the Ethics Committee of Tokushima University Hospital. The IRB guideline is equivalent to the guideline of the Ministry of Education, Culture, Sports, Science and Technology and the Ministry of Health, Labor, and Welfare in Japan.

### Patients

This retrospective study included 12 consecutive non-thymomatous MG patients (9 female and 3 male patients), who underwent extended thymectomy at Tokushima University Hospital over the last 10 years. The mean patient age was 43.3 ± 14.2 years (range, 12–59 years).

### MG assessment

MG was assessed by a neurologist (N.M.) at the time of observation, using the myasthenia gravis activities of daily living (MG-ADL) score [11]. The diagnosis of MG was made according to

the presence of clinical features mentioned in the Myasthenia Gravis Foundation of America (MGFA) clinical classification [12] and a positive result in one or more of the following tests: edrophonium test, electrophysiological test (repetitive nerve stimulation test and/or single-fiber electromyography), and test for antibodies against the AChR or muscle-specific tyrosine kinase.

Before thymectomy, all patients had been receiving anticholinesterase drugs alone or in combination with steroids. Neurologists and surgeons discussed the decision and timing for thymectomy. Of the twelve patients included in the study, eight had been receiving PSL for approximately one month before the surgery. Seven patients in this group underwent CT examination before PSL administration, while the remaining one patient underwent CT examination after PSL administration.

The PSL dose reduction extent was defined as follows:

PSL dose reduction extent = preoperative dose (maximum)–postoperative dose for 3 years (mg).

The ADL improvement rate was defined as follows:

ADL improvement rate = (preoperative ADL score–postoperative ADL score) / 24 (total ADL score).

## Surgical technique

All procedures were performed under general anesthesia. Thymectomy was performed via trans-sternal approaches. Extended thymectomy, which involved en bloc resection of the anterior mediastinal fat tissue, including the thymus, was performed according to the method reported by Masaoka et al. (we had learned this method from his co-worker Yasumasa M) [13]. Dissection was performed bluntly from the pericardium and pleura. The adipose tissues around the upper poles of the thymus, around both brachiocephalic veins, and on the pericardium were resected meticulously. If necessary, the pleural cavity was entered. The resection borders were the diaphragm caudally, thyroid gland cranially, and phrenic nerves laterally. Anticholinesterase drugs and steroids were administered according to clinical status.

Three-dimensional image analysis (Fig 1A)

We used Zio Station 2.0X software (Ziosoft Corporation, Tokyo, Japan) under the Work Station environment for thymic tissue volume assessment. One surgeon and one radiological engineer decided the extended thymectomy area. All CT scans were 1 mm slice width and were performed only at our institution (Tokushima University Hospital). They were performed following the same protocol. The automatic 3D construction of the thymus using the software is difficult because it has both high and low-density areas. Solid components indicate a high-intensity (solid) area with thymic tissues, while fatty components indicate low-density areas around the thymus with adipose tissue. The extended thymectomy area was determined after careful considerations. The basic threshold of CT value was identified using ziostation software. The thymus area was divided into solid and fat areas. In the same patient, the density of the thymus may vary even in the same CT conditions, as this study was a retrospective study, Therefore in addition to the basic CT value, we also carefully set the threshold of solid and fat density on each patient. We determined the solid and fat densities on representative slice, then traced 15–20 slices around the areas of extended thymectomy area manually (diaphragm caudally, thyroid gland orally, and phrenic nerves laterally). The ziostation software constructed 3D image of the extended thymectomy area and automatically calculated the volume of the extended thymectomy area. The total volume was considered as the volume resected by extended thymectomy (total volume = solid volume + fatty volume).

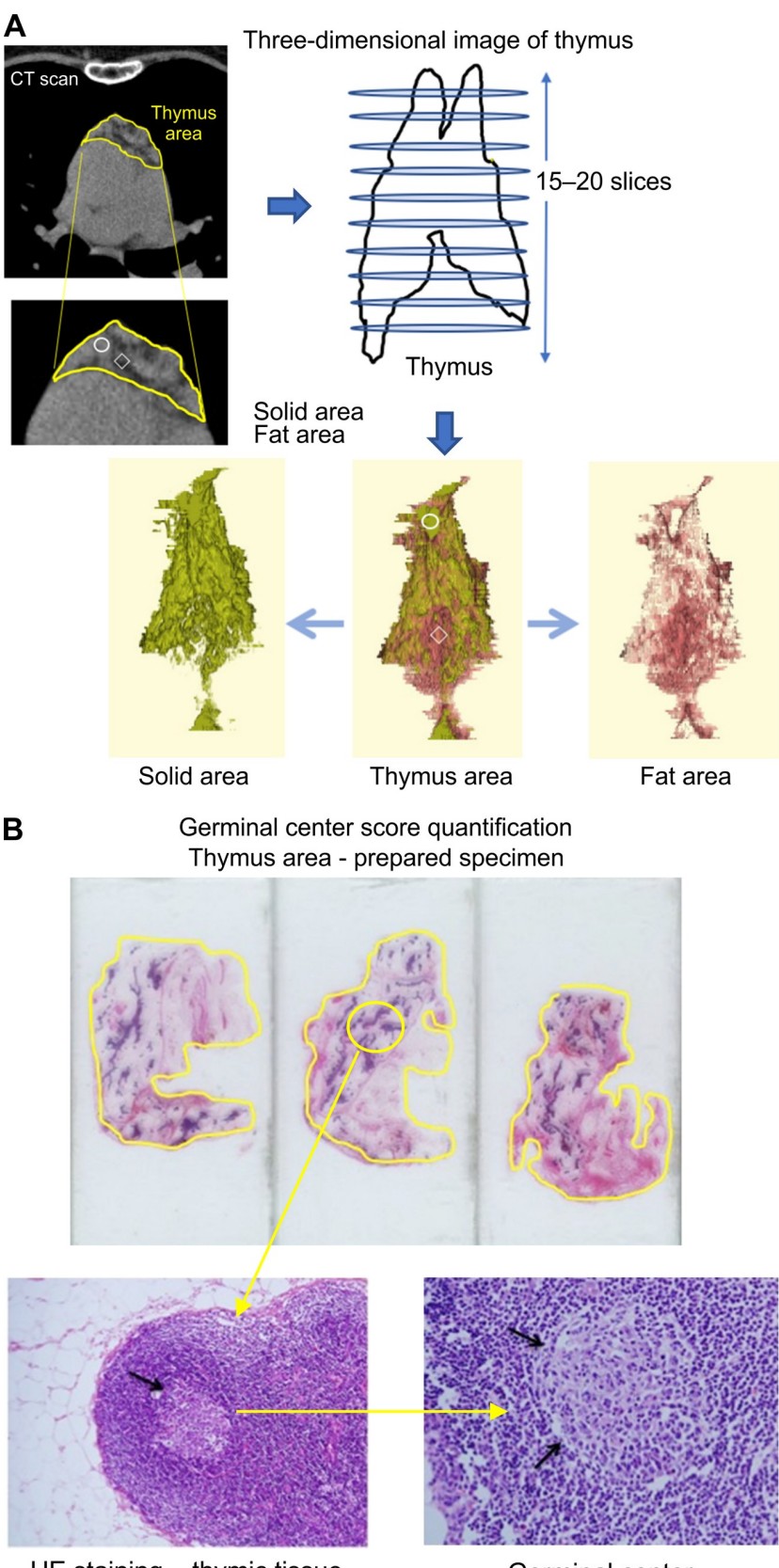

**A**

Three-dimensional image of thymus

CT scan
Thymus area

15–20 slices

Thymus

Solid area
Fat area

Solid area     Thymus area     Fat area

**B**     Germinal center score quantification
Thymus area - prepared specimen

HE staining – thymic tissue          Germinal center

**Fig 1. A.** 3D image construction and analysis of the germinal center score. The solid component indicating a high-intensity area within the thymic tissues, and the fatty component indicating a low-density area within the adipose tissue around the thymus can be seen; 3D images were constructed, and the volume of the solid area (thymic tissue) was calculated; the total volume was considered as the volume resected by extended thymectomy (total volume = solid volume + fat volume). **B.** Thymic tissues around the fatty tissues resected by extended thymectomy fixed in formalin, cut at intervals of 1 cm, and embedded in paraffin; tissue sections prepared and then stained with hematoxylin and eosin (HE) for histological examination; the number of germinal centers was counted, and the area of thymic tissue was calculated using Image J software; the number of germinal centers per area of thymic tissue (pixel) ($\times10^{-6}$) was assessed, and the highest value was considered as the germinal center score.

## Germinal center score quantification (Fig 1B)

Thymic tissues around fatty tissues resected by extended thymectomy were fixed in formalin, cut at intervals of 1 cm, and embedded in paraffin. Tissue sections were made, and they were stained with hematoxylin and eosin (HE) for histological examination. One pathologist (Y.B.) and two surgeons (M.Y. and K.K.) checked all tissue sections and counted the number of germinal centers. The area of thymic tissue was calculated using Image J software (National Institutes of Health, Bethesda, MD, USA) [14]. We assessed the number of germinal centers per area of thymic tissue (pixel) ($\times10^{-6}$) and considered the highest value as the germinal center score.

## Follow-up

Patients were followed by 2 surgeons (K.K. and M.Y.) and 1 neurologist (N.M.) every 3 months for the first year and every 6 months thereafter. Patients continued to receive medical therapy (i.e., anticholinesterase and/or steroids [PSL]). No specific immunosuppressive drug was administered. The therapeutic effect and symptomatic response to the different therapies were determined for each patient according to the MGFA postintervention status classification [15], and precise assessments were performed by a neurologist (N.M.). We defined the minimal-manifestation status as "no symptoms or functional limitations from MG, but possible presence of weakness on examination of some muscles." The steroid dose was maintained until the minimal-manifestation status was reached.

## Statistical analysis

Data are presented as mean ± standard deviation (SD). The paired t-test and correlation coefficients (Pearson and Spearman) were used to analyze the data. All statistical analyses were performed using Prism software (Graph Pad Software, La Jolla, CA). A p-value $<0.05$ was considered significant.

## Results

### Characteristics of MG patients without thymoma

The characteristics of the 12 patients MG patients without thymoma who underwent extended thymectomy are summarized in Table 1. The median age of the 12 patients without thymoma who underwent extended thymectomy was 43.3 ± 14.2. Nine of the patients were women, and 7 of the patients had class IIa/IIb MGFA classification. The mean pre- and post-surgery scores for anti AChRab and MG-ADL were 783 ± 1674.5; 155.1 ± 328, and 10.1 ± 1.7; 2.8 ± 1.9, respectively. (We evaluated the AChRAb data for 11 patients.) The difference in AChRab level was statistically significant (P$<$0.05). The mean time (months) to thymectomy was 18.9 ± 15.9."

**Table 1. Clinical characteristics of the study group.**

| | |
|---|---|
| Number of patients | 12 |
| Age, years (median ± SD) | 43.3±14.2 |
| Gender | |
| Male, n | 3 |
| female, n | 9 |
| Time to thymectomy (month± SD) | 18.9±15.9 |
| MGFA clinical classification | |
| class I | 1 |
| class IIa/IIb | 7 |
| class IIIa/IIIb | 3 |
| class IV | 0 |
| Anti-AchRAb (nmol/l) | |
| Pre-surgery (mean±SD) | 783.0±1674.5 |
| Post-surgery (mean±SD) | 155.1±328.3 |
| MG-ADL score (mean±SD) | |
| Pre-surgery | 10.1±1.7 |
| Post-surgery (1year) | 4.2 ±2.9 |
| Post-surgery (3year) | 2.8 ±1.9 |

## Pre- and post-surgery AchRab values

The mean pre-surgery and post-surgery AChRAb levels were 783.0 ± 1674.5 nmol/l and 155.1 ± 328.3 nmol/l, respectively (Fig 2A). The AChRAb level showed a significant improvement after the surgery when compared to the level before the surgery in each of the patients (paired t-test, $p < 0.05$).

The improvement rate in AChRab (calculated as 100 –Post-surgery AChRab / Pre-surgery AChRab x 100%) after PSL administration in 6 patients is illustrated in Fig 2B. There was no significant difference in the AChRab improvement rate between the pre-surgery PSL administration group (PSL(+)) and pre-surgery PSL, not administration groups (PSL(-)) (74.8±13.9 vs. 62.2±41.3 (%), p = 0.501).

Six of the seven patients in the PSL administration underwent CT examination before surgery, and there was no correlation between the AChRab improvement rate and solid volume (Spearman r = 0.21, p = 0.53) (Fig 2C).

Fig 3 illustrates the MG-ADL score (Pre-surgery, 1 year and 3-year post-surgery) for nine patients.

We identified the MG-ADL score in 9 patients. The mean preoperative, 1 year and 3-year postoperative MG-ADL scores were 10.1 ± 1.7, 4.2 ± 2.9 and 2.8 ± 1.9, respectively. The MG-ADL score showed a significant improvement between pre-surgery vs 1 year and pre-surgery vs 3 years after the operation (repeated ANOVA, p≤0.001). The MG-ADL score was not significant between 1 year and 3 years after the operation. Seven patients were administered PSL before the operation.

## Germinal center and solid volume, solid/fat ratio (Fig 4)

We evaluated the relationship between the number of germinal centers in the resected thymus and the solid volume of the thymus in pre-operative CT images. Seven patients were administered PSL before surgery (PSL(+)), six of whom underwent CT examination before CT examination. Therefore, PSL administration did not affect their Germinal center score. One patient

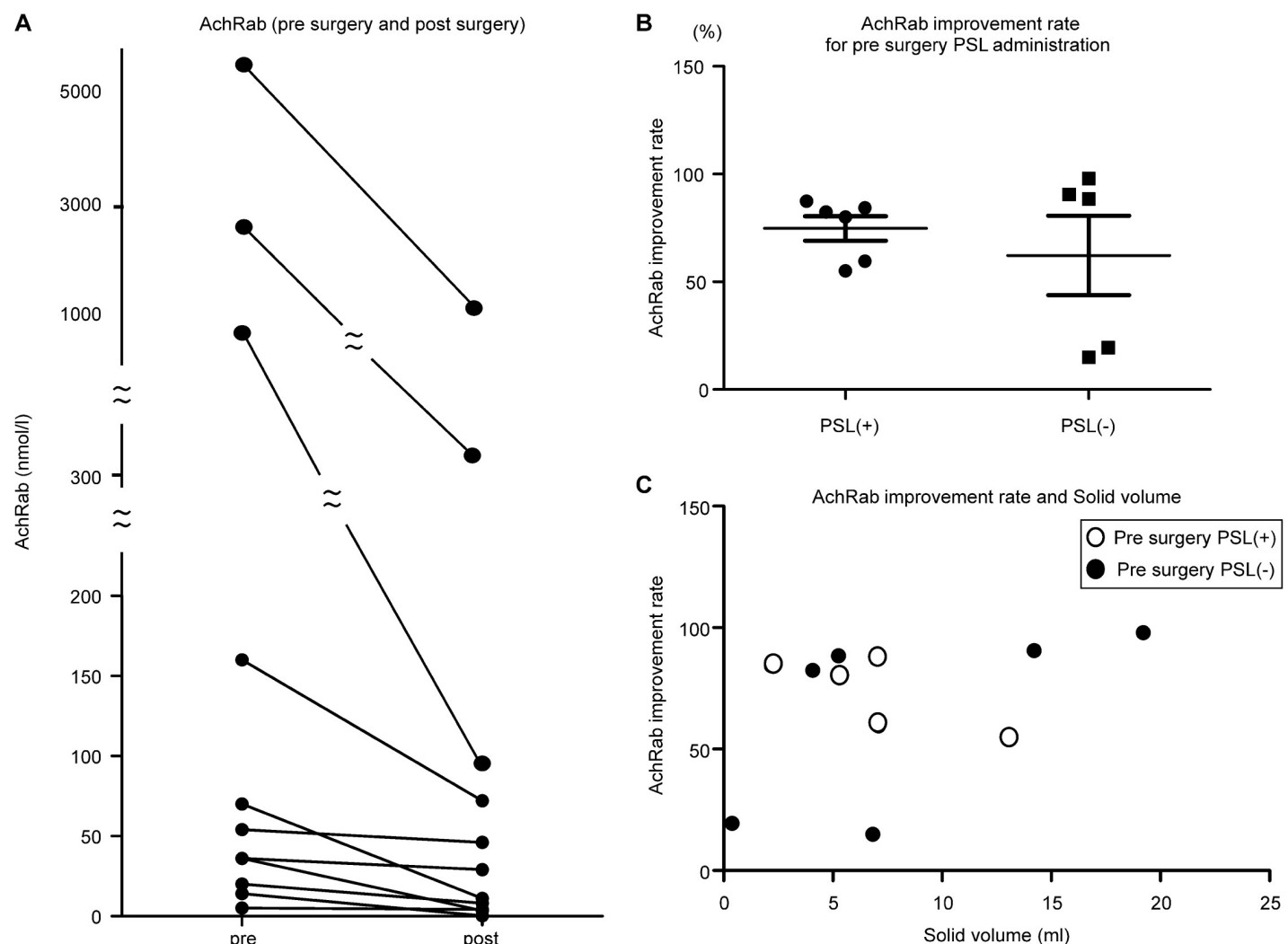

**Fig 2. A.** Pre- and post-surgery AChRab values. A significant improvement in the AChRAb levels can be seen after the operation when compared to the levels before the operation in each of the patients (paired t-test, p < 0.05). **B.** Difference between AChRab improvement rates for the pre-surgery PSL administration group (PSL(+)) and pre-surgery PSL, not administration groups (PSL(-)). Pre-surgery PSL received group was seven patients (we could not follow only one patient after surgery AChRab.); therefore, we analyzed six patients for PSL group. There is no significant difference in the AChRab between the pre- and post-surgery groups (unpaired t-test p = 0.501). **C.** Correlation between AChRab improvement rate and solid volume. There is no correlation between the AChRab improvement rate and solid volume. The white dots indicate the PSL (+) group, while the black dots denote the PSL(-) group (Spearman r = 0.21, p = 0.53).

underwent CT examination after PSL administration. We included this patient in this analysis. The number of germinal centers per unit area was strongly correlated with the solid volume (Spearman r = 0.797, p = 0.0019) (Fig 4A), and with the solid/fat ratio (Spearman r = 0.676, p = 0.0158) (Fig 4B).

Seven patients had received PSL before surgery while five patients had not. There was no significant difference between the germinal center score and pre-surgical PSL treatment (PSL (+): 351 ± 396.8, PSL(-): 519.8 ± 642.3 (x10$^{-6}$; Unpaired t-test, p = 0.585) (Fig 4C).

## pre-surgery PSL amount (mg) and Germinal center score

There is no correlation between pre-surgery PSL dose (mg) and germinal center score (Pearson r = -0.36, p = 0.424) (Fig 4D).

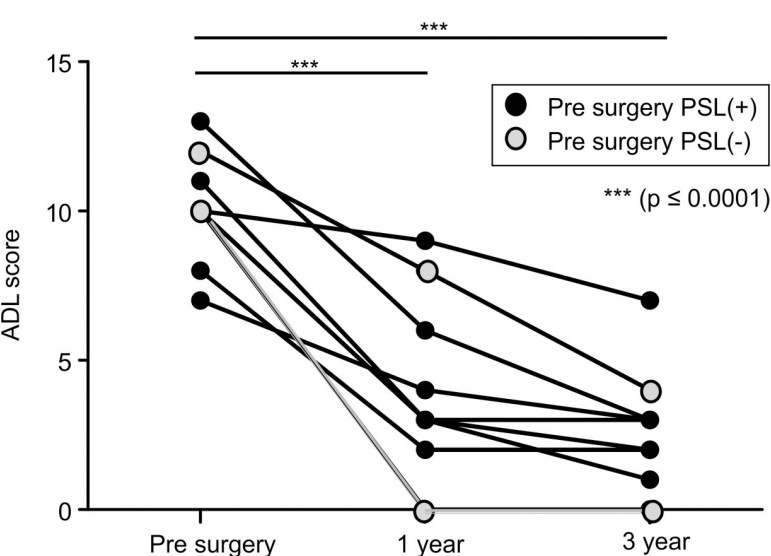

**Fig 3. MG-ADL (activities of daily living scale) score (pre-surgery and post-surgery 1 year, 3 years).** Seven patients were administered PSL preoperatively (black dots: PSL(+)). The white dots denote those who were not administered PSL preoperatively (PSL(-)).

Solid volume and MG-ADL improvement (1 and 3 years postoperatively) (Fig 5).

The ADL improvement rate was defined as follows:

ADL improvement rate = (preoperative ADL score–postoperative ADL score) / 24 (total ADL score).

The MG-ADL improvement rates at 1 year and 3 years after the surgery were not correlated with the solid volume of the thymus (one year; Pearson r = 0.398, p = 0.29, 3 years, Pearson r = 0.09, p = 0.82) (Fig 5).

The correlation between the solid volume and extent of PSL dose reduction is shown in Fig 6. One patient who received PSL after surgery was included in the analysis. There was a correlation between the extent of PSL dose reduction and solid volume of the thymus (Pearson r = 0.70, p = 0.05).

The PSL dose reduction extent was defined as follows:

PSL dose reduction extent = preoperative dose (maximum)–postoperative dose for 3 years (mg).

## Discussion

S. Berrih-Aknin et al. reviewed the role of B cells in myasthenia gravis[16]. The thymic ectopic germinal centers are sensitive to the action of the anti-inflammatory corticosteroid therapy [17]. They showed the striking changes in the size of germinal canters induced by corticosteroids in the thymus of MG patients. Germinal centers could be occasionally observed in the thymus of healthy individuals, but germinal center number and size of corticosteroid treated MG patients are very low compared with corticosteroid untreated MG patients [17].

Therefore, we must pay attention to the pre-surgery PSL treatment. We considered whether pre-operative PSL had an impact on the germinal center score of the tissue, solid volume of the CT imaging, and AChRab decrease rate. In this study, pre-surgery PSL was shown not to affect the germinal center score and AChRab decrease rate. Seven patients were administered PSL before surgery, six patients of whom underwent CT examination. Therefore, solid component

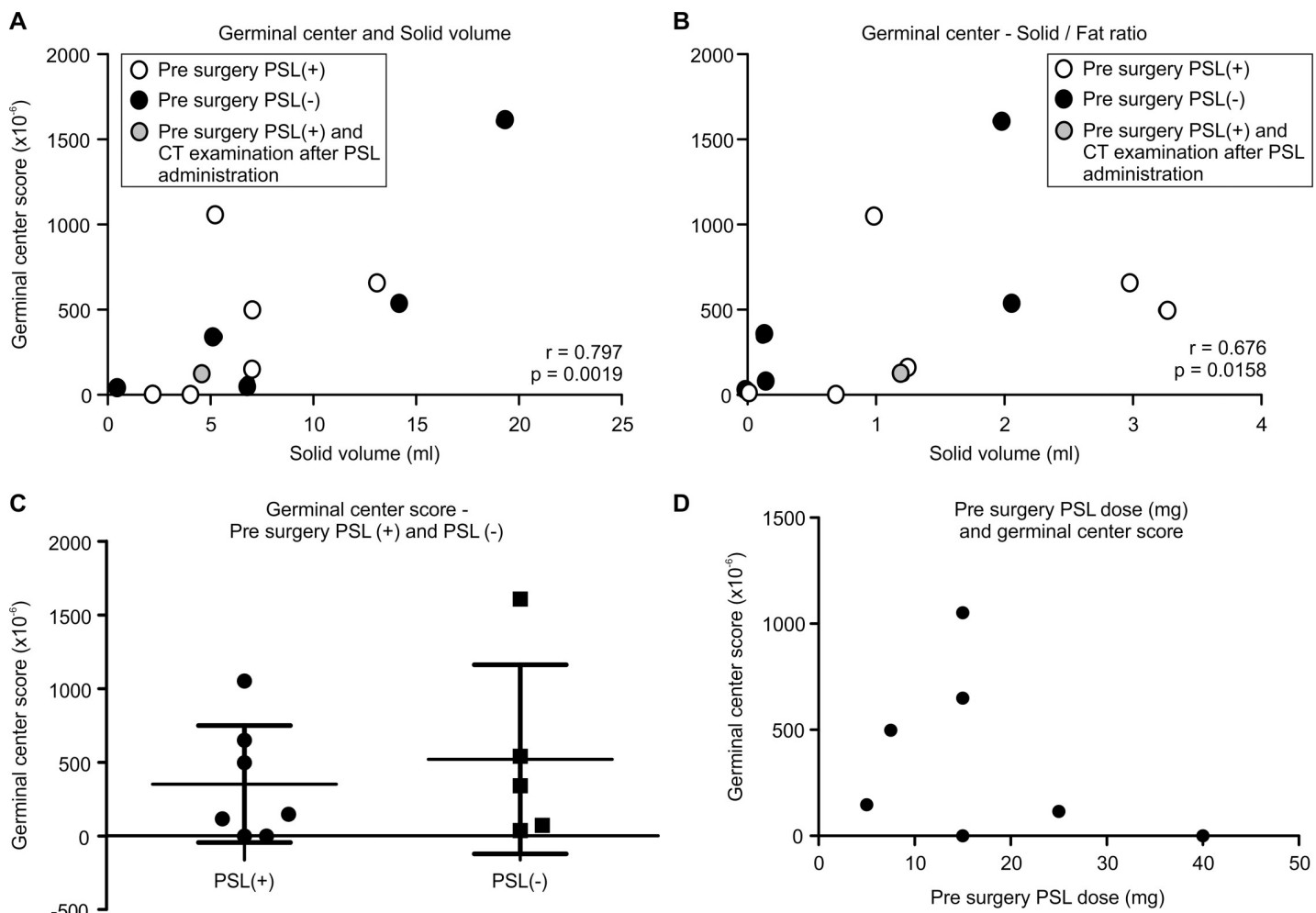

**Fig 4. A.** Germinal center and solid volume, Solid/Fat ratio. The relationship between the number of germinal centers in the resected thymus and the solid volume of the thymus is seen in preoperative CT images. **B.** Germinal center and Solid/Fat ratio. The number of germinal centers correlate with the solid/fat ratio (Spearman r = 0.676, p = 0.0158). **C.** Germinal center score and pre PSL (+) group, pre PSL (-) group. The difference in the germinal center scores between the pre PSL (+) group and pre PSL (-) group is not significant (unpaired t-test, p = 0.585). **D.** Pre-surgery PSL amount (mg) and Germinal center score. There is no correlation between the pre-surgery PSL dose (mg) and the germinal center score (Pearson r = -0.36, p = 0.424).

analysis did not affect PSL administration. Only one patient underwent CT examination after PSL administration. We included this patient in this analysis which is indicated as a gray dot (Fig 4A and 4B). Thus, preoperative PSL administration did not affect the germinal center score and AChRab decrease rate. We think the reason for the observation was due to a short PSL administration period (around one month); therefore, PSL treatment did not affect the germinal centers.

MG is an autoimmune disease of the neuromuscular junction that causes fluctuating skeletal muscle weakness. The efficacy of thymectomy for improving outcomes in patients with non-thymomatous MG is still being studied, although it has been used in clinical practice for over 60 years [2–4]. The surgical management of MG patients has improved with time and the associated morbidity and mortality are low, especially with less invasive techniques [18, 19]; however, thymectomy has risks and associated costs. Therefore, it is important to better understand its effectiveness for improving outcomes in MG patients. Extended thymectomy is considered one of the main approaches for achieving complete stable remission of MG [20–23].

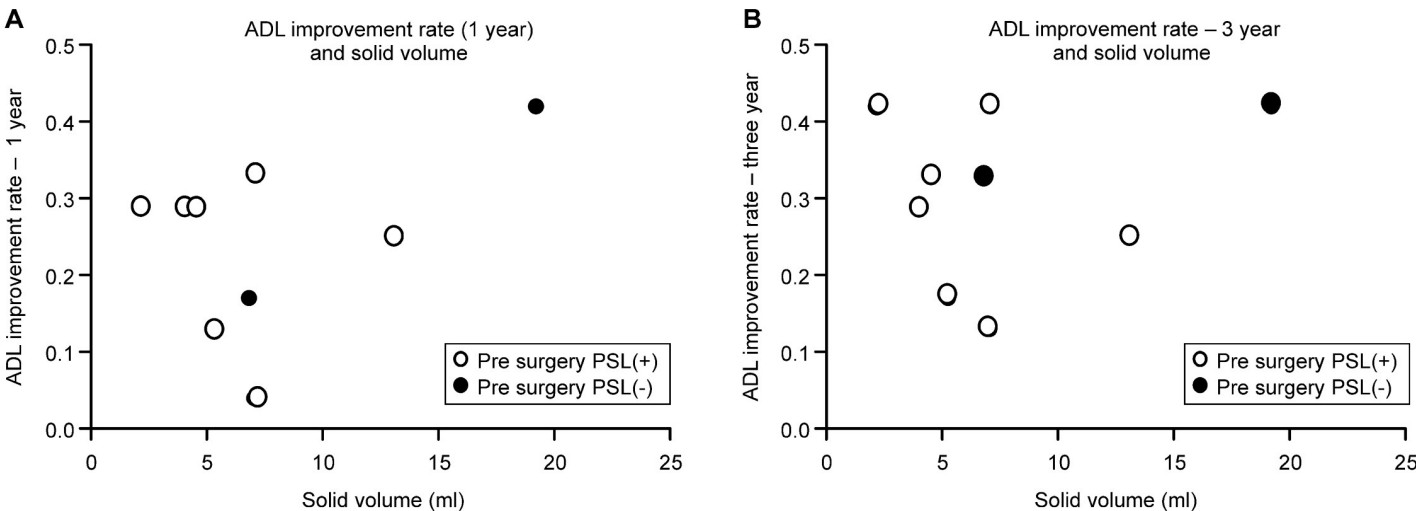

**Fig 5. A.** Solid volume and MG-ADL improvement—1 year after surgery. The MG-ADL improvement rates 1 year after the surgery do not correlate with the solid volume of the thymus (Pearson r = 0.398, p = 0.29). **B.** Solid volume and MG-ADL improvement—3 years after surgery. The MG-ADL improvement rates 3 years after the surgery do not correlate with the solid volume of the thymus (Pearson r = 0.09, p = 0.82).

The persistence of ectopic thymic tissue with germinal centers associated with AChRAb production is considered as one of the main reasons for poor outcomes after thymectomy [24–31].

In the present study, the AChRAb level significantly improved after the operation (155.1 ± 328.3 nmol/l) when compared to the level before the operation (783.0 ± 1674.5 nmol/l) in each of the patients. Additionally, the MG-ADL score significantly improved 3 years after the operation (2.8 ± 1.9) when compared to the level before the operation (10.1 ± 1.7) in each of the patients. Extended thymectomy was performed according to the original procedure

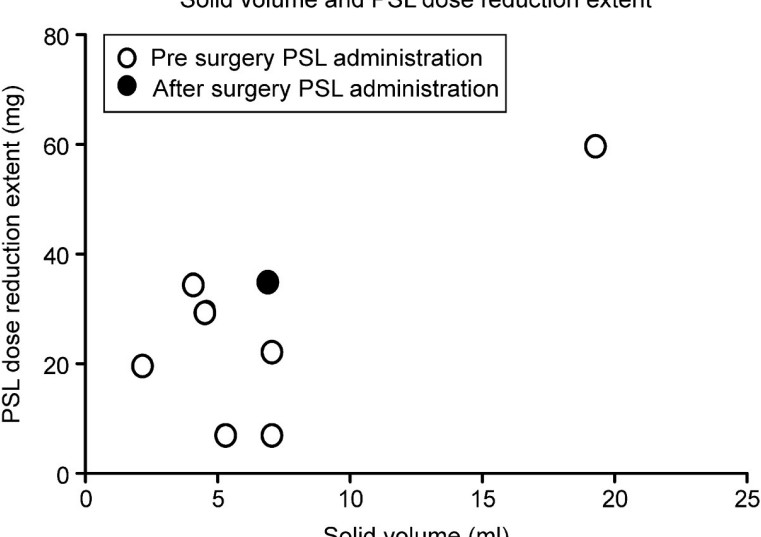

**Fig 6. Solid volume and PSL dose-reduction extent.** The PSL dose-reduction extent correlates with the solid volume of the thymus (Pearson r = 0.70, p = 0.05). One patient (black dot) was administered PSL after the surgery. We included this patient in the analysis.

presented by Masaoka et al. [13], and the serum AChRAb level was assessed at our institution. The MG-ADL score was estimated by a skilled neurologist at 1 and 3 years after the operation. We demonstrated that the AChRAb level and MG-ADL score significantly improved after extended thymectomy in our non-thymomatous MG patients. These findings are consistent with the results of the MGTX Study Group showing that the time-weighted mean quantitative myasthenia gravis score and time-weighted mean MG-ADL score over a 3-year period were better in patients who underwent thymectomy than in those who received prednisone alone (6.15 vs. 8.99, p < 0.001 and 2.24 vs. 3.41, p = 0.008, respectively) [6]. We estimated the MG symptoms of patients using only the MG-ADL score, while the previous study assessed symptoms using the quantitative myasthenia gravis score and the MG-ADL score. The findings of other studies also support our results [6, 32, 33]. Okumura et al. reported that the transition of the AChRAb level might be an index of clinical improvement in MG patients after surgery [32]. Additionally, Akihiro et al. reported that anterior mediastinal tissue volume is correlated with the AChRAb level in MG patients [33].

As hypothesized, the number of germinal centers per unit area of the resected thymic tissue was strongly correlated with the solid volume of the thymus in 3D images (Spearman r = 0.797, p = 0.0019) and with the solid/fat ratio in 3D images (Spearman r = 0.676, p = 0.016). We believe that the solid volume of the thymus and solid/fat ratio in 3D images are important predictors of extended thymectomy efficacy.

We used the MG-ADL score as an indicator of MG symptoms. We found that a greater solid volume tended to be associated with a better MG-ADL improvement at 1 year postoperatively; however, the association was not significant.

The MGTX Study Group showed that the time-weighted mean prednisone dose was significantly lower in patients who underwent thymectomy than in those who received prednisone only [6]. Therefore, extended thymectomy can help lower the steroid dose at 3 years postoperatively. Our findings are consistent with the results of the MGTX Study Group. We demonstrated that 7 patients who were administered PSL preoperatively (one patient was administered PSL post operatively) received a reduced PSL dose for 3 years after the operation and that the PSL dose reduction extent tended to correlate with the solid volume of the thymus (Fig 6). In this study, the steroid dose was maintained until the minimal-manifestation status was reached. Thus, the rate of decrease in the steroid dose reflected MG symptom improvement. Additionally, the solid volume of the thymus in 3D images can predict a decrease in the steroid dose.

## Limitations

The present study had some limitations. First, this study had a retrospective design and included a small number of patients. In the future, a prospective study should be performed with a large number of non-thymomatous MG patients undergoing extended thymectomy. Second, we could not assess the germinal centers in the entire resected thymus. We only assessed the germinal centers in the prepared thymic tissue specimens maintained in the Pathology Department of our institution. This might have resulted in pathological bias. Third, the CT imaging conditions slightly differed among the patients. Care should be taken with regard to density decisions on CT imaging of the thymus, as areas with different degrees of density (low, middle, and high density) can be identified. We decided the threshold for identification of the solid component in each patient. Additionally, 1 surgeon and 1 radiological engineer decided the extended thymectomy area manually. Therefore, the thymectomy area can be considered accurate. However, there might be limitations concerning reproducibility.

## Conclusion

We demonstrated that the AChRAb level and MG-ADL score significantly improved after extended thymectomy in non-thymomatous MG patients. The solid volume and solid/fat ratio in 3D images were strongly correlated with the number of germinal centers per unit area of the resected thymic tissue. Our findings suggest that the solid volume of the thymus can predict steroid dose reduction. Additionally, the solid volume of the thymus in pre-surgery 3D images can be the most important indicator for predicting the efficacy of extended thymectomy.

## Supporting information

**S1 Table.**
(TIF)

## Author Contributions

**Formal analysis:** Mitsuteru Yoshida.

**Investigation:** Mitsuteru Yoshida, Kazuya Kondo, Naoko Matsui, Yuishinn Izumi, Yoshimi Bando, Kouichirou Kajiura, Akira Tangoku.

**Visualization:** Michihiro Yokoishi.

**Writing – original draft:** Mitsuteru Yoshida.

**Writing – review & editing:** Mitsuteru Yoshida.

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
