## [Decision Letter · Decision Letter 0]

27 May 2020

PONE-D-20-12048

Prediction of improvement after extended thymectomy in non-thymomatous myasthenia gravis patients

PLOS ONE

Dear Dr. Yoshida,

Thank you for submitting your manuscript to PLOS ONE. After careful consideration, we feel that it has merit but does not fully meet PLOS ONE’s publication criteria as it currently stands. Therefore, we invite you to submit a revised version of the manuscript that addresses the points raised during the review process.

Please indicate the correlation of the antibody decrease with the volumes, include results  on the effect of steroid treatment before the surgery and provide further data on the evaluation of the CT scans. Also please include the limitations raised by the reviewers in the revised manuscript. 

The figure 1 is not understandable and the figures 2 and b and 3 a and b could be combined.

We look forward to receiving your revised manuscript.

Kind regards,

Güher Saruhan-Direskeneli, M.D.

Academic Editor

PLOS ONE

2. In the ethics statement in the manuscript and in the online submission form, please provide additional information about the patient records used in your retrospective study. Specifically, please ensure that you have discussed whether the IRB or ethics committee waived the requirement for informed consent. If patients provided informed written consent to have data from their medical records used in research, please include this information.

Reviewers' comments:

Reviewer's Responses to Questions

**Comments to the Author**

1. Is the manuscript technically sound, and do the data support the conclusions?

Reviewer #1: Yes

Reviewer #2: Yes

2. Has the statistical analysis been performed appropriately and rigorously? 

Reviewer #1: I Don't Know

Reviewer #2: Yes

3. Have the authors made all data underlying the findings in their manuscript fully available?

Reviewer #1: Yes

Reviewer #2: Yes

4. Is the manuscript presented in an intelligible fashion and written in standard English?

Reviewer #1: Yes

Reviewer #2: Yes

5. Review Comments to the Author

Reviewer #1: This retrospective study included 12 consecutive non-thymomatous MG patients, who underwent extended thymectomy over the last 10 years. The study assessed the number of germinal centers per unit area, change in the serum AChRAb level, postoperative MG improvement, PSL dose reduction extent, and solid volume of the thymus.

The present study aimed to determine whether the solid volume of the thymus calculated using three-dimensional (3D) imaging could be used to predict the efficacy of thymectomy.

The number of germinal centers per unit area was significantly correlated with the solid volume of the thymus (R = 0.58, p = 0.004). The PSL dose reduction extent tended to be correlated with the solid volume. Findings in this study suggest that the solid volume of the thymus can possibly predict steroid dose reduction. Additionally, the solid volume of the thymus in 3D images is the most important indicator for predicting the efficacy of extended thymectomy.

1- The AChRAb improvement rate was not correlated with the solid volume of the thymus.

2- All 8 patients who had been administered PSL preoperatively received a reduced PSL dose after the operation. To make the study homogenous I would prefer including patients without PSL. In this study, if you exclude those patients there would not be enough patients to study. Yet, we do not have a solid information if preoperative PSL has positive effects on postoperative condition. This is one of the most important negativity of this study. Besides, as the patients have PSL before they have their preopartive CT, germinal centers would have been effected too.

3- The MG-ADL improvement rates at 1 and 3 years postoperatively were not correlated with the solid volume of the thymus.

4- The number of germinal centers per unit area was significantly correlated with the solid volume,

5- We need to know other antibodies in the population.

6- Findings suggest that A:the solid volume of the thymus can possibly predict steroid dose reduction. B: the solid volume of the thymus in 3D images is the most important indicator for predicting the efficacy of extended thymectomy.

With limited number of patients, long duration, retrospective design, inhomogeneous MG treatment, I would not be comfortable with the conclusion and findings. Yet, we know that extended thymectomy is good for patients with generalized MG with a positive AChRAb.

The study is scientific and inspiring. It could be a nice guide for those who have high number of patients, who could build uniform patient populations. I would increase the limitations of the study including recommendations from this review.

Reviewer #2: In this article, Yoshida et al correlated preoperative CT scans and 3D reconstructions of the thymus gland with postoperative findings (lymph follicles/germinal centers) in patients with myasthenia gravis. The authors found a good correlation between the solid thymic areas in 3D scans and the number of lymphoid follicles and clinical benefit from thymectomy. Although this study is retrospective and based on a relatively small number of patients and will thus require further independent confirmation, the clinical implications are potentially important and may help in the management of these patients.

Specific comments:

This reviewer is not a radiologist, but to me the specifications in the materials and methods sections on the CT analyses require more detail. Were all the CT scans from a single institution? Were they performed under similar conditions protocols? Were they high resolution CT? This should be specified. In case the CT scans were from different institutions, the authors should give general recommendations on how the CT should be performed and whether specific protocols should be followed.

Did the authors observe differences between patients who received steroids vs. patients who did not (solid areas on CT, number of lymphoid follicles)? The case numbers are probably too low to draw any conclusions, but this should be briefly discussed.

6. PLOS authors have the option to publish the peer review history of their article (what does this mean?). If published, this will include your full peer review and any attached files.

Reviewer #1: Yes: Alper Toker, WVU, Morgantown, US

Reviewer #2: No

---

## [Author Response · Author response to Decision Letter 0]

28 Aug 2020

Dear specific reviewer

Thank you for the concerns regarding the inadequacy in describing the specifications of the CT analysis, which we have now adequately described.

Thank you for the concerns regarding the effect for pre-surgery PSL administration. We discussed about it precisely. We considered the information above and added the following sentence.

Could you check the file : Response to Reviewers.

Thank you so much.

Best regards,

Mitsuteru Yoshida

---

## [Editor Report · Decision Letter 1]

14 Sep 2020

Prediction of improvement after extended thymectomy in non-thymomatous myasthenia gravis patients

PONE-D-20-12048R1

Dear Dr. Yoshida,

We’re pleased to inform you that your manuscript has been judged scientifically suitable for publication and will be formally accepted for publication once it meets all outstanding technical requirements.

Kind regards,

Güher Saruhan-Direskeneli, M.D.

Academic Editor

PLOS ONE

Additional Editor Comments (optional):

A minor point:

Acetylcholine receptor should be abbreviated as "AChR" throughout the manuscript.
---

## [Editor Report · Acceptance letter]

18 Sep 2020

PONE-D-20-12048R1 

Prediction of improvement after extended thymectomy in non-thymomatous myasthenia gravis patients 

Dear Dr. Yoshida:

I'm pleased to inform you that your manuscript has been deemed suitable for publication in PLOS ONE. Congratulations! Your manuscript is now with our production department. 

Kind regards, 

on behalf of

Dr. Güher Saruhan-Direskeneli 

Academic Editor

PLOS ONE